# Side Effects of Kaolin and Bunch-Zone Leaf Removal on Generalist Predators in Vineyards

**DOI:** 10.3390/insects14020126

**Published:** 2023-01-25

**Authors:** Elena Cargnus, Federico Tacoli, Francesco Boscutti, Pietro Zandigiacomo, Francesco Pavan

**Affiliations:** Department of Agricultural, Food, Environmental and Animal Sciences, University of Udine, Via delle Scienze 206, 31000 Udine, Italy

**Keywords:** Araneae, cultural control, entomophagous arthropods, functional biodiversity, IPM, species richness, spiders

## Abstract

**Simple Summary:**

Kaolin application and bunch-zone leaf removal are two cultural practices that contribute to the control of main pests in vineyards, such as leafhoppers and the European grapevine moth. In the context of Integrated Pest Management (IPM), a two-year study on the side effects of these practices on generalist predators was conducted in three vineyards of a grape-growing area in north-eastern Italy. This study provides the first data on the influence of kaolin and bunch-zone leaf removal on spiders and predatory insects occurring in vineyards. It was demonstrated that moderate use of kaolin has a negligible impact on generalist predatory arthropods in vineyards, while no negative effects were associated with bunch-zone leaf removal. Therefore, these practices were compatible with IPM strategies.

**Abstract:**

In vineyards, kaolin application and bunch-zone leaf removal (LR) were effective in the control of leafhoppers and *Lobesia botrana*, but their side effects on generalist predators are still poorly understood. In north-eastern Italian vineyards, the impact of kaolin and LR on species and functional diversity of spiders, as well as the abundance of spiders and generalist predatory insects, was assessed in one vineyard for two consecutive years and in two vineyards for one year. The ecological indices of the spider community were never influenced by kaolin and only in one case were they influenced by LR. At the spider family level, kaolin reduced the abundance of Araneidae, Oxypidae and Salticidae, but only in single cases. In single cases, kaolin reduced the amount of *Orius* sp. anthocorids and increased that of Scymninae coccinellids, whereas LR increased the amount of *Aeolothrips* sp. The moderate use of kaolin and the application of LR had negligible and inconsistent impacts on generalist predatory arthropods in vineyards and were therefore, compatible with IPM strategies.

## 1. Introduction

Plants sprayed with kaolin, a white aluminum silicate, become visually, tactually and chemically unrecognizable to arthropod pests [1]. The treated substrates impaired the adhesion of a Pentatomidae bug and a Tephritidae fly due to particles of kaolin sticking to their legs [2]. Pests can be negatively affected by kaolin mostly by reducing egg laying and feeding activity [3]. Its applications were effective in pest control in orchards and field crops [4,5,6,7,8].

In vineyards, kaolin was effective against sap-sucking pests, such as the leafhoppers *Hebata vitis* (Göthe) [9] [syn. *Empoasca vitis* (Göthe)], *Zygina rhamni* Ferrari, *Arboridia kermanshah* Dlabola, *Erasmoneura vulnerata* (Fitch) and *Scaphoideus titanus* Ball [10,11,12,13], the grape phylloxera *Daktulosphaira vitifoliae* (Fitch) [14] and the vine cicada *Psalmocharias alhageos* [15], as well as carpophagous pests, such as the European grapevine moth *Lobesia botrana* (Denis and Schiffermüller) [16,17] and the spotted wing drosophila *Drosophila suzukii* Matsumura [18]. Kaolin provided an effective control of two polyphagous scarab beetles *Macrodactylus subspinosus* (Fabricius) and *Popillia japonica* Newman, feeding on grapevine leaves and bunches [19,20].

Bunch-zone leaf removal is a common cultural practice in vineyards, which consists of removing all the leaves that cover bunches, contributing to the control of both bunch rots and *L. botrana* [21,22]. Larvae of the carpophagous generations can die just after hatching due to high temperatures recorded on the sun-exposed berries where eggs were laid [23]. The combination of kaolin with bunch-zone leaf removal provides synergistic control of *L. botrana* with the added benefit that kaolin reduces sunburn damage to the berries often caused by sun exposure of bunches [17,24].

In the context of Integrated Pest Management (IPM), kaolin and bunch-zone leaf removal can be considered a valid alternative to synthetic insecticides only if their side effects on natural enemies are comparably less negative and not such as to favour the resurgence of pests. In fact, spiders and generalist predatory insects can have a significant impact on pest populations in several agro-ecosystems [25,26,27]. The ability to feed on alternative prey allows generalists to also be present inside a crop when pest population densities are still low, and they may delay or prevent pest outbreaks [28]. In vineyards, spiders and generalist predatory insects form diverse communities [29,30,31] and can contribute to the control of vineyard pests [32,33,34,35,36]. The richness of predators can be negatively influenced by cultural practices and pesticide applications while also being favoured by the presence of surrounding groves and permanent inter-row vegetation [37,38,39,40,41].

In several studies, no negative effects of kaolin were reported for spiders (Araneae), Heteroptera and Coleoptera Coccinellidae both in orchards and cotton [42,43,44,45,46]. However, kaolin reduced the abundance of these generalist predatory taxa in other studies conducted in orchards [5,6,47,48,49].

The effects of kaolin on non-target arthropods in vineyards are still poorly understood. Two consecutive kaolin applications did not affect the parasitisation of *H. vitis* and *Z. rhamni* eggs by *Anagrus atomus* (L.) [11], whereas kaolin was moderately harmful to the phytoseiid mites *Typhlodromus pyri* Scheuten and *Kampimodromus aberrans* (Oudemans) both in the field and in laboratory trials [50]. The adoption of bunch-zone leaf removal did not substantially affect phytoseiid populations in vineyards [50]. However, no literature data are available to determine if this cultural practice could negatively affect other generalist predators as a result of changes that occur in the structure and microclimate of the grapevine canopy [23]. Therefore, in some vineyards, the effects of kaolin and bunch-zone leaf removal on the diversity and amount of spiders and generalist predatory insects were studied.

## 2. Materials and Methods

### 2.1. Study Vineyards

The study was conducted in 2015 and 2016 in three vineyards located in a flat grape-growing area of north-eastern Italy (Gorizia district, Friuli Venezia Giulia region), in the framework of trials against *H. vitis* and *Z. rhamni* [11] and *L. botrana* [17], herein named Vineyard A (2015 and 2016), Vineyard B (2015) and Vineyard C (2016). Vineyard A (45°57′51″ N, 13°26′49″ E, 56 m a.s.l., cultivar Pinot Gris) was a 10-year-old conventional vineyard, with grapevines grown using the Guyot training system and distanced between and along rows of 2.5 m and 0.8 m, respectively. Vineyard B (45°57′20″ N, 13°26′50″ E, 50 m a.s.l., cultivar Pinot Gris) was a 30-year-old organic vineyard with grapevines grown using the double-arched Guyot training system and distanced between and along rows of 2.8 m and 1 m, respectively. Vineyard C (45°58′02″ N, 13°31′31″ E, 53 m a.s.l., cultivar Pinot Gris) was a 15-year-old organic vineyard with grapevines grown using the Guyot training system and distanced between and along rows of 2.2 m and 0.7 m, respectively. In the three vineyards, herbaceous vegetation on the inter-rows was present. During the two-year study, standard fungicide programs were followed and insecticides were not applied. In terms of fungicides, in Vineyards B and C, only copper and sulphur products were used, whereas in Vineyard A, synthetic substances were also sprayed.

### 2.2. Experimental Design

In all vineyards, the effects of kaolin and bunch-zone leaf removal (LR) were studied and, only in Vineyard A, they were evaluated for two consecutive years. Kaolin (Surround WP, Tessenderlo Kerley Inc., Phoenix, AZ, USA, 2% suspension) was sprayed at the rate of 1000 L/ha two times a year, except on Vineyard A in 2016 where three applications occurred because the first was notably washed off by rain [16] (Table 1). Kaolin was applied at the occurrence of defined *L. botrana* phenological stages as previously reported [17]. All applications were performed using a backpack sprayer (Oleo-Mac, Sp-126, Emak S.p.A., Bagnolo in Piano RE, Italy).

In all vineyards, a randomised block design with four replicates, each corresponding to a different grapevine row, was adopted. Blocks were divided into two plots (kaolin and control), each consisting of 28 (Vineyard A) or 20 (Vineyard B) or 24 (Vineyard C) grapevines. In half of the grapevines of each plot, leaves were manually removed so to obtain two subplots, with and without LR (LR and no LR) (Table 1). In Vineyard A, the plots sprayed with kaolin and subplots subjected to LR were the same in the two years.

### 2.3. Samplings

In 2015 and 2016, spider amount was assessed by a drop cloth method (in 2015 in both Vineyards A and B: 11, 22, 29 June, and 6 July; in 2016 in Vineyard A: 6, 20, 28 June, and 6 July; in 2016 in Vineyard C: 7, 21 June, and 1, 8 July). Only in 2015, both spiders and generalist predatory insects were monitored using yellow sticky traps that were replaced weekly (Vineyards A and B, from 4 June to 18 August). Yellow sticky traps were used in 2015 because they are an effective sampling method for leafhoppers which were considered only during this year [11]. For both sampling methods, the first sampling was carried out before the first kaolin application. The drop cloth method consisted of manually shaking a grapevine canopy five times, grabbing the apical part of the trunk, and collecting fallen spiders on a pale cloth sheet (65 × 45 cm). At each sampling date, the spiders collected on 10 grapevines in the central part of each subplot were separately preserved in vials with 70% aqueous ethanol. Yellow sticky traps (20 × 10 cm) were smeared with glue (Temo-O-Cid^®^, Kollant Srl, Vigonovo, VE, Italy) and hung on the horizontal wires of the grapevine trellis at about 1.5 m from the ground level to be inside the canopy, but not covered by leaves. At each sampling date, one trap per subplot was installed.

Spiders and generalist predatory insects were observed in the laboratory under a dissecting microscope. Spiders collected with the drop cloth method were identified to different taxonomic levels, i.e., adults were assigned to genus or species and juvenile individuals to family or genus. In addition, they were grouped on the basis of hunting strategy (i.e., web-builders and hunters, with the latter further divided into ambushers and active hunters), and body size (i.e., ≤4 mm, small; 4.1–7 mm, medium; 7.1–10 mm, large). The distance from the front of the carapace, without the chelicerae, to the end of the abdomen was measured. Among predatory insects captured by yellow sticky traps, the following taxa were identified: *Aeolothrips* sp. (Thysanoptera: Aeolothripidae), Scymninae (Coleoptera: Coccinellidae), and Coccinellidae non-Scymninae, *Chrysoperla carnea* s.l. (Stephens) (Neuroptera: Chrysopidae), and *Orius* sp. (Hemiptera: Anthocoridae). Among spiders captured by yellow sticky traps, web-builders were distinguished from hunters, which were separately counted at the family level.

### 2.4. Statistical Analyses

The effects of kaolin and LR on the spider communities collected with the drop cloth method in the three vineyards were examined in terms of species diversity (Shannon index H’), three functional diversity (FD) indices [i.e., functional richness (width of a niche space filled with species), functional evenness (the evenness of the distribution of species abundance in niche space) and functional distance (mean of individual species to the centroid of all species in the community)] and functional guilds (i.e., body size and hunter percentage) [52,53]. All indices were calculated using the FD package [53,54], which implements a distance-based approach that allows for the analysis of both continuous and categorical variables. The data relative to H’ and FD indices, and functional guilds for spiders (collected with drop cloth) were compared using an ANOVA test applied to linear mixed-effects models (LMM) [55]. For these analyses, spider community data of Vineyard A were pooled at the year level to highlight whether the effect of kaolin and LR recorded in each year was consistent. For the same comparisons, the data collected in the two Vineyards B and C were pooled to highlight whether the effects of kaolin and LR were consistent. The sampling date was included in the models as a random factor. Model assumptions were verified by visual inspection of diagnostic plots. When assumptions were violated, a logarithmic transformation was applied. Changes in spider species assemblage among all factors were inspected using a multivariate approach. A Kruskal’s non-metric multidimensional scaling (NMDS) unconstrained ordination [56,57] was performed with the Bray-Curtis (dis)similarity index, two dimensions (k = 2) on Wisconsin and square root transformed data, separately for Vineyard A (stress = 0.11) and Vineyards B and C (stress = 0.14). All the multivariate analyses were performed using the “vegan” R package [58].

To evaluate the effects of kaolin and LR on the number of spiders and predatory insects (collected with a drop cloth and yellow sticky traps), for each vineyard and year, an ANOVA test applied to linear mixed-effects models (LMM) followed by Tukey’s post hoc test was used [59]. The sampling date was included in the models as a random factor. Model assumptions were verified by visual inspection of diagnostic plots. When assumptions were violated, a logarithmic transformation was applied.

Within spiders, the relative proportion of hunting-strategy groups (hunters and web-builders) or subgroups (ambushers and active hunters) in each vineyard was compared using a G test of goodness of fit. To compare percentage data, a Fisher’s exact test or Ryan’s test [60] were used when two or more proportions were considered, respectively.

Statistical analyses were performed with SPSS version 20 [61] or R software version R 4.0.4 [62].

## 3. Results

### 3.1. Spiders and Generalist Predatory Insects in Vineyards

#### 3.1.1. Spiders by the Drop Cloth Method

Spider populations were more than twice as abundant in Vineyard B (N. 160) compared to the other vineyards (A 2015: N. 70; A 2016: N. 65; C: N. 60). In whole, spiders belonged to nine families (the web-builders Araneidae, Linyphiidae and Theridiidae; the ambushers Thomisidae; and the active hunters Gnaphosidae, Miturgidae, Oxyopidae, Salticidae and Sparassidae) and 28 different taxa (Appendix A). The highest diversity in the number of families was observed in 2016 in Vineyard A (nine out of nine), but no less than seven families were observed in the other cases (i.e., Vineyard A and B in 2015 and Vineyard C in 2016).

Based on the hunting strategy, hunters were more represented than web-builders with significant differences at the G test in Vineyard A (in 2015: 76%, *p* < 0.0001; in 2016: 72%, *p* < 0.0001) and in Vineyard B (62%, *p* < 0.001), but not in Vineyard C (58%, *p* = 0.13). The proportion of web-builders was significantly higher in Vineyards B and C than in Vineyard A at 0.05 level (Ryan’s test).

Among web-builders, Araneidae were prevalent in all vineyards (Vineyard A in 2015: 71%, *p* = 0.056; Vineyard A in 2016: 61%, *p* = 0.28; Vineyard B: 52%, *p* = 0.97; Vineyard C: 68%, *p* = 0.044), with a prevalence of *Nuctenea* sp. in Vineyard A and Vineyard C, and *Mangora acalypha* (Walckenaer) in Vineyard B. In total, Linyphiidae represented around a third of web-builders.

Among hunters, the subgroup “ambushers”, exclusively represented by Thomisidae, was on average prevalent on the subgroup “active hunters” in Vineyard A and Vineyard B, even if the difference did not reach the significance level at the G test (Vineyard A in 2015: 58%, *p* = 0.15; Vineyard A in 2016: 57%, *p* = 0.23; Vineyard B: 54%, *p* = 0.45), whereas the subgroup “active hunters” was significantly prevalent in Vineyard C (80%, *p* < 0.0001). Among ambushers (i.e., Thomisidae), *Xysticus* sp. (60%) was the prevalent taxon. Among active hunters, Oxyopidae of the genus *Oxyopes* (40%) and Salticidae (36%) were the more represented.

#### 3.1.2. Predatory Arthropods by Yellow Sticky Traps

In both Vineyards A and B, the amount of captured predatory insects belonging to the considered taxa was prevalent on spiders (84% and 16% of capture, respectively) (Appendix A). Among the predatory insects, *Aeolothrips* sp. was the most abundant (85% in Vineyard A, and 71% in Vineyard B). The number of captured Coccinellidae in Vineyard B was three times higher than in Vineyard A.

Total spiders were more abundant in Vineyard B than in Vineyard A, but the proportion of web-builders and hunters was the same (40% and 60%, respectively) (Appendix A). Among hunters, Thomisidae were the most abundant family in both vineyards (around 51%).

### 3.2. Effect of Kaolin and Bunch-Zone Leaf Removal on Predatory Insects and Spiders

#### 3.2.1. Vineyard A

During the two years under study (2015 and 2016), spiders collected by the drop cloth method were not affected by kaolin and LR in terms of H’ and FD indices and functional guilds (i.e., body size and hunter percentage) (Table 2). The interactions among factors were not significant, suggesting a lack of a cumulative effect on the spider community. However, a sensitive shift in the spider community was observed for year, kaolin and LR (Figure 1a). NMDS1 discriminated mainly between 2015 and 2016 and partially between LR and no LR subplots, while NMDS2 discriminated mainly between kaolin and the control. Kaolin (NMDS2) was associated with a higher abundance of *Xysticus* sp., while *Oxyopes* sp. was more frequent in the control.

Significant differences in spider amount occurred only at the family level, i.e., for active hunters Oxyopidae in 2015 and Salticidae in 2016, and for web-builders Araneidae in 2016 (Table 3). No significant differences were observed for LR compared to no LR.

Among predatory insects captured by yellow sticky traps in 2015, kaolin significantly increased Scymninae (Table 4). On the contrary, *Orius* sp. showed lower captures in the kaolin than in the control, but not at a significant level. *Aeolothrips* sp. showed significantly higher captures in LR than no LR. Spiders captured by yellow sticky traps were not significantly affected by either kaolin or LR. Concerning spider families, Thomisidae showed higher captures in LR than no LR, but not at a significant level.

#### 3.2.2. Vineyards B and C

In neither of the two vineyards were spiders collected by the drop cloth method affected by kaolin in terms of H’ and FD indices and considered functional guilds (i.e., body size and hunter percentage) (Table 5). On other hand, LR significantly decreased H’ and functional evenness (Table 5), mostly in Vineyard C when combined with kaolin (Figure 2). The comparison of the two vineyards showed significantly higher H’, functional richness and evenness in Vineyard B compared to Vineyard C (Table 5 and Figure 2). NMDS analysis showed that the two vineyards had different spider communities (NMDS1, Figure 1b). Kaolin did not influence the spider community, while LR produced a significant change in the spider community shown along the NMDS2. In particular, LR favoured the abundance of *Xysticus* sp., while no LR resulted in a higher occurrence of *Oxyopes lineatus* Latreille.

The total spiders collected by the drop cloth method in both vineyards were not significantly affected by kaolin and LR, not even when considering hunting-strategy groups or families (Table 6). The proportion of the active hunters on the total hunters [i.e., ambushers (Thomisidae) plus active hunters] was not changed by kaolin in both Vineyards B and C (Vineyard B: from 57% to 40%, *p* = 0.23; Vineyard C: constant 86%, *p* = 1.0).

Among the predatory insects captured by yellow sticky traps in vineyard B, lower captures of *Orius* sp. in the kaolin than in the control were recorded (Table 7). Predatory insects were not influenced by LR. Spiders were not significantly affected by either kaolin or LR.

#### 3.2.3. Comparison between the Two Sampling Methods

In 2015, in Vineyard A, a significantly higher percentage of web-builders than hunters was associated with yellow sticky traps compared to the drop cloth method (42% vs. 24%, *p* = 0.007, at Fisher’s exact test). This occurrence was not confirmed for Vineyard B (38% vs. 40%, *p* = 0.766, at Fisher’s exact test). Among hunters, a significantly higher percentage of Salticidae was captured with yellow sticky traps than by the drop cloth method (Vineyard A: 34% vs. 9%, *p* = 0.0007, at Fisher’s exact test; Vineyard B: 23% vs. 11%, *p* = 0.017).

## 4. Discussion

### 4.1. Spiders and Generalist Predatory Insects in Vineyards

The same taxa of spiders found in the vineyards of the present study are reported as predators of *L. botrana* larvae, pupae and adults in other studies [34,63,64,65]. For example, *Salticus scenicus* (Clerck) (Salticidae), recorded in the present study, has been reported feeding on *H. vitis* nymphs in French vineyards, although its contribution to leafhopper control has not been considered important [66]. In Portuguese vineyards, as well as in the present study, *H. vitis* dead adults were found on spider webs [65]. *Oxyopes lineatus* (Oxyopidae), whose individuals were observed preying on *H. vitis* adults during the present study, has been reported feeding on Cicadellidae [67]. Even Linyphiidae, found in low numbers in the present study, could have a role in the control of *H. vitis*, as the species *Bathyphantes pallidus* (Banks) was found to be a valuable predator of the leafhopper *Empoasca fabae* (Harris) in a field of alfalfa in Kentucky (USA) [68]. Moreover, in American vineyards, Salticidae and Theridiidae have been reported as effective predators of *Erythroneura* spp. leafhoppers [69], even if plant vigour or other factors that favour *Anagrus* spp. egg parasitoids, seem to be more important than spider abundance [32,33,70]. Finally, species of *Cheiracanthium* sp. (Miturgidae), also found in two vineyards in the present study, are widespread predators of *Erythroneura* spp. leafhoppers in Californian vineyards [35,36,38,71].

Among web-builders, the prevalence of Araneidae in the present study is in agreement with other studies [30,72]. Their higher abundance in both Vineyards B and C compared to Vineyard A could be associated with the synthetic insecticides applied in the previous years in this last vineyard [73,74]. To this purpose, under laboratory conditions, it was observed that the web of *Araniella* sp. (Araneidae) did not protect its owner from insecticide spraying [75]. Nevertheless, in an apple orchard, web builders (Theridiidae) resulted in being less susceptible to insecticide spraying than hunters probably because their three-dimensional webs protected them from droplets [76].

The comparison of yellow sticky traps and drop cloth to monitor spiders in vineyards was performed in the present study and in that of Nobre et al. [77]. Both studies are in agreement with the higher amount of web-builders captured by yellow sticky traps than the drop cloth method. Additionally, among hunters, a relatively higher proportion of Salticidae captured by yellow sticky traps compared with the drop cloth method was observed in both studies. When the canopy is shaken, the Salticidae individuals could jump outside the surface area of the pale cloth sheet resulting in an underestimation of their number. Moreover, according to Nobre et al. [77], the moving of Salticidae along leaves and wires could increase their interception by traps. In agreement with Nobre et al. [77], we can state that the yellow sticky traps are not sufficiently representative of the spider fauna in vineyards since they alter the effective proportion among spider families. Furthermore, the drop cloth method is more suitable for species identification.

The five taxa of generalist predatory insects captured by yellow sticky traps (i.e., *Aeolothrips* sp., Scymninae, Coccinellidae non-Scymninae, *Orius* sp. and *C. carnea* s.l.) are reported feeding on grapevine pests. *Aeolothrips* sp. are predators of phytophagous thrips in vineyards [78]. Scymninae presence in vineyards is often associated with mealybug infestation [79]. *Chrysoperla carnea* s.l. is reported as a predator of *L. botrana* larvae [63,65] and motile forms of *H. vitis* and *Z. rhamni* [80]. The generalist pirate bugs *Orius* spp. are effective control agents of tetranychid and eriophyid mites in vineyards [29], but can also feed on *H. vitis* [81]. In that regard, a strong correlation between *H. vitis* and *Orius niger* (Wolff) population densities was observed using yellow sticky traps, even if the predatory activity could not be confirmed visually [82]. In Californian vineyards, low densities of leafhoppers and thrips were attributed to the impacts of *Orius* spp. and coccinellids (see Miles et al. [39]).

### 4.2. Influence of Kaolin and Bunch-Zone Leaf Removal on Spiders and Generalist Predatory Insects

In this study, kaolin affected the spider communities in terms of abundance only in three cases recorded in Vineyard A, i.e., the Oxyopidae and Salticidae active hunters, and the Araneidae web-builders. The prevalence of cases with negative kaolin effects on hunters compared to web-builders agrees with studies conducted in apple orchards [44,47]. Concerning the web builders, a higher mortality of *Araniella cucurbitina* (Clerck) was observed under laboratory conditions after the ingestion of great amounts of silk covered with kaolin [83], it is because Araneidae often repair or rebuild their webs by ingesting them [25]. This feeding behaviour can explain why in our study Araneidae appeared to be more sensitive to kaolin than the other web-builder spiders.

In the present study, the relatively reduced negative effects of kaolin on spiders can be explained by the low number (two or three) of kaolin applications (20 kg/ha per application). In the literature, negative effects on spiders were sometimes observed with a higher number of kaolin applications than in the present study (total amount per hectare 5–10 times higher) [6,42,44,46,47]. In apple orchards, after 12 kaolin applications (45 kg/ha per application), spider densities were significantly reduced [47], and at more than four weeks from the last application the abundance of spiders was still significantly lower in the kaolin-treated plots [6]. On the other hand, in a cotton field, spider densities were not reduced after several weekly kaolin applications (24–50 kg/ha per application) [46]. In an apple orchard, a negative effect of kaolin on spiders was observed only in the second year after four yearly applications (60 kg/ha per application) [44]. Even in a pecan orchard, after seven and six kaolin applications (50 kg/ha per application) in the first and second year, respectively, spiders became fewer on kaolin-treated plots in the second year [42]. In Vineyard A of the present study, a reduction in the abundance of Araneidae and Salticidae was observed only in the second of two consecutive years of application of kaolin on the same plots, even though the species diversity of spiders increased in comparison to the first year. After all, the heavy reduction of *H. vitis* and *Z. rhamni* populations in plots sprayed with kaolin [11,17] could have reduced the availability of prey for hunters causing their dispersion. The absence of negative effects on Thomisidae, and even an average increase in their populations observed in some cases, could be explained by the reduction of intra-guild predation by the active hunters when they disperse [27,84,85,86].

Two kaolin applications interfered with the abundance of predatory insects only in two cases: *Orius* sp. decreased and Scymininae increased. With the same number of kaolin applications, a negative effect on *Orius* sp. was also reported in olive orchards [5]. For another minute pirate bug, *Anthocoris nemoralis* (Fabricius) (Anthocoridae), negative effects of kaolin were found under laboratory and field conditions [5,87], but this occurrence was not confirmed by another laboratory study [88]. In the present study, the lower population of *Orius* sp. could also be an indirect effect of the reduction of leafhopper populations by kaolin [11], as anthocorids can have a role in *H. vitis* control [81,82]. In Vineyard A, the higher densities of Scymninae observed in the kaolin compared to the control plots could be explained because kaolin can favour mealybug abundance [89,90].

In the present study, kaolin did not affect Coccinellidae non-Scymninae in agreement with other studies [42,43,46], even if negative effects have been reported in some cases [5,44]. However, in the laboratory, kaolin was harmless to these Coccinellidae [87,91].

Additionally, in the present study, *C. carnea* s.l. was not negatively influenced by kaolin, although the data reported in the literature did not always confirm the absence of negative effects [45,92]. However, in the laboratory kaolin was harmless to *C. carnea* and *Chrysoperla externa* (Hagen) [87,88,93]. Kaolin did not negatively affect *Aeolothrips* sp., whose individuals spend most of their life feeding on small arthropods and pollen occurring in herbaceous vegetation of the inter-row.

Bunch-zone leaf removal influenced captures of spiders and predatory insects only in two cases: *Aeolothrips* sp. that were more abundant in LR of Vineyard A and spiders that showed lower species diversity and functional evenness in Vineyard C. The increase in captures of *Aeolothrips* can be explained by the preference of Thysanoptera for yellow sticky traps exposed to sunlight [94,95] as occurred for traps placed on grapevines submitted to LR.

Regarding the effect of LR on the spider community, which significantly reduced spider diversity in one of the three vineyards, opposing suggestions were reported in the literature. The study of Pennington et al. [96], in which grapevines grown with minimal or intensive pruning were compared, did not show the negative effects of a more reduced canopy. On the contrary, a higher number of hunters and web-builders was attributed to the abundant foliage on apple trees [97].

## 5. Conclusions

In vineyards, moderate use of kaolin and the application of LR have shown minor impact on spiders and generalist predatory insects; therefore, they are suitable cultural practices in the context of IPM strategies. The negative effects are also negligible after two consecutive years of kaolin applications on the same plots. Even the few cases of a decrease in hunter spiders could be a consequence of the negative effects of kaolin on leafhoppers, which are the most frequent prey of spiders inhabiting grapevine canopy. Regardless, considering the positive effects of kaolin and LR on the control of grapevine key pests, the overall outcome of their adoption is positive. The use of these cultural practices in the context of IPM reduces the risk that pests exceed their economic thresholds and, thus, the need for insecticide applications in vineyard.

## Figures and Tables

**Figure 1 insects-14-00126-f001:**
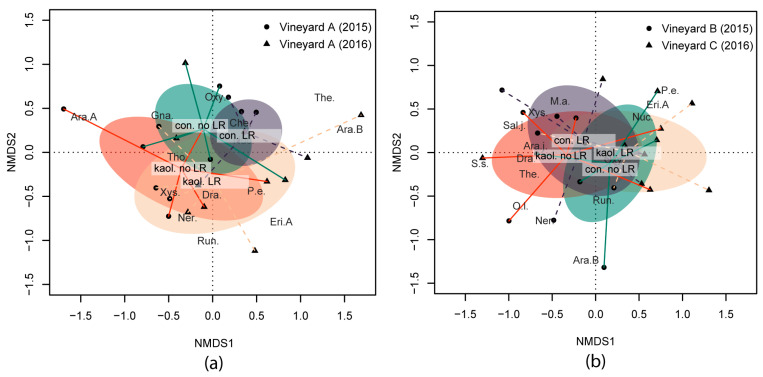
Biplot of the non-metric multidimensional scaling (NMDS) ordination applied to spider communities collected by the drop cloth method in (**a**) Vineyard A (2015 and 2016); and in (**b**) Vineyards B (2015) and C (2016), respectively. Shifts in spider community under the different combinations of kaolin application and bunch-zone leaf removal (LR) are indicated by the centroids of each combination (i.e., con. no LR = control without LR, con. LR = control with LR, kaol. no LR = kaolin without LR, kaol. LR = kaolin with LR) and the standard error of the average of scores (shaded elliptic area with 95% confidence limit). A selection of species was plotted according to species scores with a further abundance priority selection. Species’ full names are reported in Appendix A.

**Figure 2 insects-14-00126-f002:**
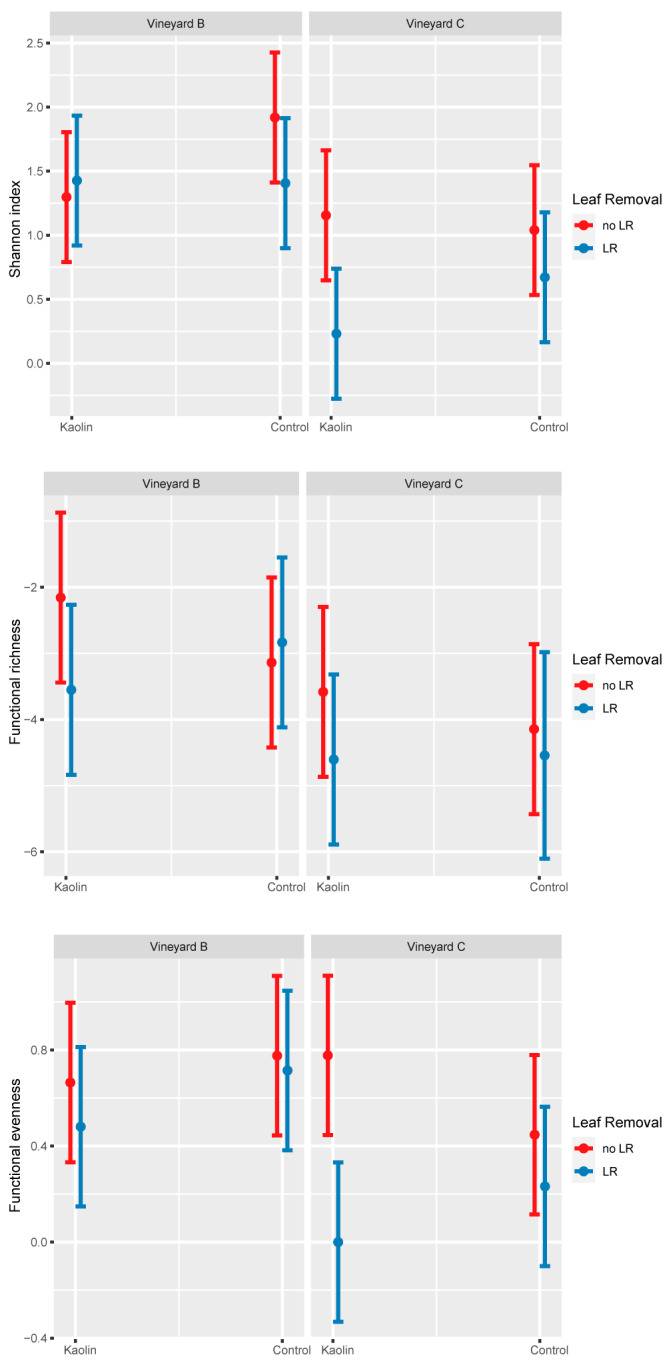
Spiders collected by the drop cloth method in Vineyards B and C. Trends of spider diversity (Shannon index), and functional richness and evenness. No significant interactions emerged during the statistical analysis. No LR = without leaf removal, LR = with leaf removal.

**Table 1 insects-14-00126-t001:** Application timings of kaolin and bunch-zone leaf removal (LR), and sampling dates in 2015 for Vineyards A and B, and in 2016 for Vineyards A and C. Phenological growth stages of grapevine in BBCH-scale are reported [51].

Date	Timing	BBCH
Vineyards A and B—2015		
17 June	LR	73
18 June	First kaolin spraying	75
24 June	Second kaolin spraying	75
Vineyard A—2016		
10 June	LR and First kaolin spraying	71
11–19 June	Rain: about 90 mm in 8 days http://www.osmer.fvg.it/archivio.php (accessed on 20 June 2016)	71–74
20 June	Second kaolin spraying	74
28 June	Third kaolin spraying	77
Vineyard C—2016		
10 June	LR and First kaolin spraying	71
24 June	Second kaolin spraying	77

**Table 2 insects-14-00126-t002:** Spiders collected by the drop cloth method in Vineyard A (2015 and 2016). Effects of kaolin, bunch-zone leaf removal (LR) and time (year) and their interactions on spider diversity and functional guilds (i.e., body size and hunter percentage). The outcomes rely on the ANOVA test applied to linear mixed-effects models (LMM). Significant differences are highlighted in bold.

Indices and Functional Guilds	Factors	F	D.F.	*p*
Species diversity (Shannon index H’)	Intercept	**63.305**	**1, 14**	**<0.0001**
	kaolin	1.790	1, 14	0.202
	LR	0.536	1, 14	0.476
	year	0.039	1, 14	0.847
	kaolin:LR	0.611	1, 14	0.448
	kaolin:year	0.050	1, 14	0.826
	year:LR	0.594	1, 14	0.454
	kaolin:LR:year	0.485	1, 14	0.498
Functional richness	Intercept	**64.774**	**1, 14**	**<0.0001**
	kaolin	1.307	1, 14	0.272
	LR	0.001	1, 14	0.982
	year	4.024	1, 14	0.065
	kaolin:LR	0.907	1, 14	0.357
	kaolin:year	0.007	1, 14	0.934
	year:LR	0.681	1, 14	0.423
	kaolin:LR:year	1.398	1, 14	0.257
Functional evenness	Intercept	**25.068**	**1, 14**	**0.000**
	kaolin	1.982	1, 14	0.181
	LR	0.139	1, 14	0.715
	year	0.587	1, 14	0.456
	kaolin:LR	4.024	1, 14	0.065
	kaolin:year	0.017	1, 14	0.899
	year:LR	0.097	1, 14	0.760
	kaolin:LR:year	0.676	1, 14	0.425
Functional distance	Intercept	**45.589**	**1, 13**	**<0.0001**
	kaolin	0.272	1, 13	0.611
	LR	0.048	1, 13	0.830
	year	0.100	1, 13	0.757
	kaolin:LR	1.426	1, 13	0.254
	kaolin:year	1.841	1, 13	0.198
	year:LR	0.288	1, 13	0.600
	kaolin:LR:year	0.288	1, 13	0.601
Body size	Intercept	**92.915**	**1, 13**	**<0.0001**
	kaolin	0.181	1, 13	0.677
	LR	1.039	1, 13	0.327
	year	0.857	1, 13	0.372
	kaolin:LR	0.952	1, 13	0.347
	kaolin:year	0.516	1, 13	0.485
	year:LR	0.822	1, 13	0.381
	kaolin:LR:year	0.135	1, 13	0.719
log (hunter % + 0.1)	Intercept	4.178	1, 13	0.062
	kaolin	1.313	1, 13	0.273
	LR	0.001	1, 13	0.975
	year	0.405	1, 13	0.536
	kaolin:LR	0.094	1, 13	0.764
	kaolin:year	0.026	1, 13	0.873
	year:LR	1.613	1, 13	0.226
	kaolin:LR:year	2.174	1, 13	0.164

**Table 3 insects-14-00126-t003:** Spiders collected by the drop cloth method in Vineyard A in each of the two sampling years (2015 and 2016). For total spiders, different groups (web-builders and hunters) and families, the average number of captures per replicate throughout the sampling period (mean ± standard error) is reported. The results of ANOVA test applied to linear mixed-effects models for kaolin and bunch-zone leaf removal (LR) factors are reported. When the number of spiders collected over the whole sampling period was less than 10, no statistical analyses were carried out. Significant differences are highlighted in bold.

Taxon	Kaolin					LR				
	Control	Kaolin	F	D.F.	*p*	No LR	LR	F	D.F.	*p*
2015										
Total spiders	5.0 ± 1.4	3.8 ± 1.4	2.308	1, 12	0.155	4.4 ± 1.4	4.4 ± 1.4	0.000	1, 12	1.000
Web-builders	1.1 ± 1.0	1.0 ± 0.7	0.034	1, 12	0.857	1.3 ± 1.0	0.9 ± 0.7	0.303	1, 12	0.592
Araneidae	0.9 ± 0.9	0.6 ± 0.6	0.509	1, 12	0.489	0.9 ± 0.6	0.6 ± 0.6	0.057	1, 12	0.816
Linyphiidae	0.1 ± 0.3	0.1 ± 0.3	−	−	−	0.3 ± 0.3	0.0 ± 0.0	−	−	−
Theridiidae	0.1 ± 0.3	0.3 ± 0.3	−	−	−	0.1 ± 0.3	0.3 ± 0.3	−	−	−
Hunters	3.9 ± 1.3	2.8 ± 1.4	3.522	1, 12	0.085	3.1 ± 1.3	3.5 ± 1.4	0.391	1, 12	0.543
Thomisidae	1.8 ± 1.0	2.1 ± 1.2	0.370	1, 12	0.554	1.9 ± 1.1	2.0 ± 1.1	0.041	1, 12	0.843
Gnaphosidae	0.3 ± 0.3	0.3 ± 0.3	−			0.3 ± 0.3	0.3 ± 0.3	−	−	−
Oxyopidae	1.5 ± 1.1	0.1 ± 0.3	**6.153**	**1, 12**	**0.029**	0.5 ± 0.5	1.1 ± 1.0	1.271	1, 12	0.282
Salticidae	0.4 ± 0.4	0.3 ± 0.5	−	−	−	0.5 ± 0.6	0.1 ± 0.3	−	−	−
2016										
Total spiders	4.4 ± 1.6	3.8 ± 1.6	0.311	1, 12	0.587	3.3 ± 1.6	4.9 ± 1.6	2.104	1, 12	0.173
Web-builders	1.4 ± 0.9	0.9 ± 0.7	0.889	1, 12	0.364	0.9 ± 0.8	1.4 ± 0.8	0.889	1, 12	0.364
Araneidae	1.1 ± 0.6	0.3 ± 0.3	**7.000**	**1, 12**	**0.021**	0.5 ± 0.5	0.9 ± 0.6	1.286	1, 12	0.279
Linyphiidae	0.3 ± 0.3	0.5 ± 0.5	−	−	−	0.4 ± 0.4	0.4 ± 0.4	−	−	−
Theridiidae	0.0 ± 0.0	0.1 ± 0.3	−	−	−	0.0 ± 0.0	0.1 ± 0.3	−	−	−
Hunters	3.0 ± 1.6	2.9 ± 1.7	0.014	1, 12	0.908	2.4 ± 1.4	3.5 ± 1.8	1.120	1, 12	0.311
Thomisidae	1.1 ± 1.0	2.3 ± 1.4	2.505	1, 12	0.139	1.1 ± 0.8	2.3 ± 1.5	2.505	1, 12	0.139
Gnaphosidae	0.1 ± 0.3	0.3 ± 0.3	−	−	−	0.1 ± 0.3	0.3 ± 0.3	−	−	−
Miturgidae	0.1 ± 0.3	0.0 ± 0.0	−	−	−	0.0 ± 0.0	0.1 ± 0.3	−	−	−
Oxyopidae	0.3 ± 0.3	0.0 ± 0.0	−	−	−	0.1 ± 0.3	0.1 ± 0.3	−	−	−
Salticidae	1.3 ± 0.8	0.3 ± 0.3	**5.647**	**1, 12**	**0.035**	0.9 ± 0.7	0.6 ± 0.5	0.063	1, 12	0.563
Sparassidae	0.1 ± 0.3	0.1 ± 0.3	−	−	−	0.1 ± 0.3	0.1 ± 0.3	−	−	−

**Table 4 insects-14-00126-t004:** Insects and spiders captured with yellow sticky traps in Vineyard A in 2015. For total spiders and their groups (web-builders and hunters), and taxa of insects and spiders, the average number of captures per trap throughout the sampling period (mean ± standard error) is reported. The results of ANOVA test applied to linear mixed-effects models for kaolin and bunch-zone leaf removal (LR) factors are reported. Significant differences are highlighted in bold.

Taxon	Kaolin					LR				
	Control	Kaolin	F	D.F.	*p*	No LR	LR	F	D.F.	*p*
Insects										
*Aeolothrips* sp.	59.6 ± 10.5	71.0 ± 20.2	0.444	1, 12	0.518	38.0 ± 8.4	92.6 ± 15.6	**10.230**	**1, 12**	**0.008**
Scymninae	3.1 ± 0.5	6.4 ± 1.0	**8.380**	**1, 12**	**0.013**	3.9 ± 10.9	5.6 ± 1.1	2.430	1, 12	0.145
Coccinellidae non Scymninae	3.9 ± 0.6	2.8 ± 0.6	1.827	1, 12	0.201	2.6 ± 0.6	4.0 ± 0.6	2.729	1, 12	0.124
*Chrysoperla carnea* s.l.	2.3 ± 0.69	1.0 ± 0.4	3.000	1, 12	0.109	1.8 ± 0.4	1.5 ± 0.7	0.120	1, 12	0.735
*Orius* sp.	2.6 ± 0.8	1.0 ± 0.3	3.596	1, 12	0.082	2.4 ± 0.7	1.3 ± 0.6	1.723	1, 12	0.214
Spiders										
Total	15.8 ± 2.5	12.5 ± 1.4	1.257	1, 12	0.284	12.4 ± 21.5	15.9 ± 2.4	1,457	1, 12	0.251
Web-builders	6.6 ± 1.3	5.0 ± 0.5	1.317	1, 12	0.274	4.8 ±0.5	6.9 ± 1.3	2.252	1, 12	0.159
Hunters	9.1 ± 1.4	7.5 ± 1.2	0.719	1, 12	0.413	7.6 ± 1.3	9.0 ± 1.3	0.515	1, 12	0.487
Thomisidae	4.9 ± 1.0	3.6 ± 0.7	1.181	1, 12	0.298	3.1 ± 0.7	5.4 ± 1.9	3.827	1, 12	0.074
Gnaphosidae	0.4 ± 0.3	0.3 ± 0.2	0.176	1, 12	0.682	0.1 ± 0.1	0.5 ± 0.3	1.588	1, 12	0.232
Oxyopidae	0.5 ± 0.3	0.9 ± 0.4	0.574	1, 12	0.463	0.5 ± 0.3	0.9 ± 0.4	0.574	1, 12	0.463
Salticidae	3.4 ± 0.9	2.0 ± 0.8	1.435	1, 12	0.254	3.4 ± 0.9	2.0 ± 0.7	1.435	1, 12	0.254

**Table 5 insects-14-00126-t005:** Spiders collected by the drop cloth method in the two Vineyards B (2015) and C (2016). Effects of kaolin, bunch-zone leaf removal (LR) and vineyard and their interactions on spider diversity and functional guilds (i.e., body size and hunter percentage). The outcomes rely on the ANOVA test applied to linear mixed-effects models. Significant differences are highlighted in bold.

Indices and Functional Guilds	Factors	F	D.F.	*p*
Species diversity (Shannon index H’)	Intercept	**176.940**	**1, 14**	**<0.0001**
	kaolin	1.880	1, 14	0.192
	LR	**6.171**	**1, 14**	**0.026**
	year	**19.129**	**1, 14**	**0.001**
	kaolin:LR	0.016	1, 14	0.902
	kaolin:year	0.167	1, 14	0.689
	year:LR	1.811	1, 14	0.200
	kaolin:LR:year	3.143	1, 14	0.098
Functional richness	Intercept	**155.792**	**1, 13**	**<0.0001**
	kaolin	0.068	1, 13	0.798
	LR	1.903	1, 13	0.191
	year	**9.958**	**1, 13**	**0.008**
	kaolin:LR	2.115	1, 13	0.170
	kaolin:year	0.008	1, 13	0.928
	year:LR	0.025	1, 13	0.877
	kaolin:LR:year	0.413	1, 13	0.532
Functional evenness	Intercept	**85.379**	**1, 14**	**<0.0001**
	kaolin	0.312	1, 14	0.585
	LR	**7.818**	**1, 14**	**0.014**
	year	**7.094**	**1, 14**	**0.019**
	kaolin:LR	2.389	1, 14	0.145
	kaolin:year	1.006	1, 14	0.333
	year:LR	2.845	1, 14	0.114
	kaolin:LR:year	0.985	1, 14	0.338
Functional distance	Intercept	**133.834**	**1, 13**	**<0.0001**
	kaolin	0.224	1, 13	0.644
	LR	2.945	1, 13	0.110
	year	4.493	1, 13	0.054
	kaolin:LR	0.038	1, 13	0.849
	kaolin:year	0.013	1, 13	0.913
	year:LR	0.566	1, 13	0.465
	kaolin:LR:year	0.377	1, 13	0.550
Body size	Intercept	**172.178**	**1, 13**	**<0.0001**
	kaolin	1.332	1, 13	0.269
	LR	3.646	1, 13	0.079
	year	0.838	1, 13	0.377
	kaolin:LR	0.002	1, 13	0.968
	kaolin:year	0.270	1, 13	0.612
	year:LR	0.441	1, 13	0.518
	kaolin:LR:year	0.022	1, 13	0.886
log (hunter % + 0.1)	Intercept	**18.915**	**1, 13**	**0.001**
	kaolin	0.055	1, 13	0.818
	LR	0.154	1, 13	0.701
	year	2.916	1, 13	0.112
	kaolin:LR	1.762	1, 13	0.207
	kaolin:year	0.483	1, 13	0.499
	year:LR	2.805	1, 13	0.118
	kaolin:LR:year	1.408	1, 13	0.257

**Table 6 insects-14-00126-t006:** Spiders collected by the drop cloth method in the two Vineyards B (2015) and C (2016). For total spiders, different groups (web-builders and hunters) and families, the average number of captures per replicate throughout the sampling period (mean ± standard error) is reported. The results of ANOVA test applied to linear mixed-effects models for kaolin and bunch-zone leaf removal (LR) factors are reported. When the number of spiders collected over the whole sampling period was less than 10, no statistical analyses were carried out.

Taxon	Kaolin					LR				
	Control	Kaolin	F	D.F.	*p*	No LR	LR	F	D.F.	*p*
Vineyard B										
Total spiders	9.9 ± 2.4	10.1 ± 3.0	0.029	1, 12	0.868	10.4 ± 32.4	9.6 ± 3.0	0.261	1, 12	0.619
Web-builders	3.8 ± 1.4	3.9 ± 2.0	0.014	1, 12	0.907	4.1 ± 1.8	3.5 ± 1.7	0.352	1, 12	0.564
Araneidae	2.5 ± 1.3	1.5 ± 1.2	0.591	1, 12	0.457	2.3 ± 1.2	1.8 ± 1.2	0.024	1, 12	0.880
Linyphiidae	0.9 ± 0.8	1.6 ± 1.5	0.591	1, 12	0.457	1.3 ± 1.4	1.3 ± 1.0	0.000	1, 12	1.000
Theridiidae	0.4 ± 0.4	0.8 ± 0.8	0.000	1, 12	1.000	0.6 ± 0.6	0.5 ± 0.6	0.667	1, 12	0.430
Hunters	6.1 ± 2.1	6.3 ± 2.2	0.009	1, 12	0.926	6.3 ± 2.1	6.1 ± 2.3	0.009	1, 12	0.926
Thomisidae	2.9 ± 1.7	3.8 ± 2.2	0.452	1, 12	0.514	3.1 ± 1.7	3.5 ± 2.1	0.083	1, 12	0.778
Gnaphosidae	0.5 ± 0.7	1.0 ± 0.9	0.558	1, 12	0.469	0.9 ± 0.8	0.6 ± 0.8	0.140	1, 12	0.715
Oxyopidae	1.4 ± 1.1	1.0 ± 0.7	0.252	1, 12	0.625	1.0 ± 0.8	1.4 ± 1.1	0.252	1, 12	0.625
Salticidae	0.9 ± 0.9	0.5 ± 0.5	0.325	1, 12	0.579	0.8 ± 10.8	0.6 ± 0.6	0.036	1, 12	0.852
Sparassidae	0.5 ± 0.6	0.0 ± 0.0	−	−	−	0.5 ± 0.6	0.0 ± 0.0	−	−	−
Vineyard C										
Total spiders	4.4 ± 1.2	3.1 ± 1.4	2.793	1, 12	0.121	3.9 ± 1.3	3.6 ± 1.3	0.310	1, 12	0.588
Web-builders	1.8 ± 0.9	1.4 ± 0.9	0.474	1, 12	0.504	1.5 ± 0.9	1.6 ± 0.9	0.053	1, 12	0.822
Araneidae	1.3 ± 0.8	0.9 ± 0.5	0.692	1, 12	0.422	1.0 ± 0.6	1.1 ± 0.7	0.077	1, 12	0.786
Linyphiidae	0.5 ± 0.6	0.5 ± 0.5	−	−	−	0.5 ± 0.5	0.5 ± 0.6	−	−	−
Hunters	2.6 ± 1.0	1.8 ± 0.9	1.862	1, 12	0.197	2.4 ± 1.0	2.0 ± 1.0	0.828	1, 12	0.381
Thomisidae	0.4 ± 0.4	0.3 ± 0.3	−	−	−	0.1 ± 0.3	0.5 ± 0.5	−	−	−
Gnaphosidae	0.0 ± 0.0	0.1 ± 0.3	−	−	−	0.1 ± 0.3	0.0 ± 0.0	−	−	−
Miturgidae	0.1 ± 0.3	0.0 ± 0.0	−	−	−	0.0 ± 0.0	0.1 ± 0.3	−	−	−
Oxyopidae	1.0 ± 0.7	0.6 ± 0.5	1.080	1, 12	0.319	0.9 ± 0.6	0.8 ± 0.7	0.120	1, 12	0.735
Salticidae	1.1 ± 0.7	0.8 ± 0.8	0.176	1, 12	0.682	1.3 ± 1.0	0.6 ± 0.5	1.588	1, 12	0.232

**Table 7 insects-14-00126-t007:** Insects and spiders captured with yellow sticky traps in Vineyard B in 2015. For total spiders and their groups (web-builders and hunters), and taxa of insects and spiders, the average number of captures per trap throughout the sampling period (mean ± standard error) is reported. The results of ANOVA test applied to linear mixed-effects models for kaolin and bunch-zone leaf removal (LR) factors are reported. Significant differences are highlighted in bold.

Taxon	Kaolin					LR				
	Control	Kaolin	F	D.F.	*p*	No LR	LR	F	D.F.	*p*
Insects										
*Aeolothrips* sp.	55.6 ± 12.4	92.5 ± 21.2	2.020	1, 12	0.181	64.5 ± 19.0	83.6 ± 17.7	0.543	1, 12	0.475
Scymninae	5.9 ± 6.9	9.5 ± 8.7	2.422	1, 12	0.146	16.3 ± 3.1	22.1 ± 5.2	1.061	1, 12	0.323
Coccinellidae non-Scymninae	6.8 ± 1.4	8.4 ± 1.3	0.610	1, 12	0.450	7.3 ± 1.6	7.9 ± 1.2	0.090	1, 12	0.769
*Chrysoperla carnea* s.l.	2.6 ± 0.9	2.0 ± 0.5	0.316	1, 12	0.584	1.9 ± 0.8	2.8 ± 0.6	0.620	1, 12	0.446
*Orius* sp.	3.0 ± 0.8	0.9 ± 0.4	**5.070**	**1, 12**	**0.044**	2.1 ± 0.9	1.8 ± 0.5	0.158	1, 12	0.698
Spiders										
Total spiders	19.4 ± 2.8	20.8 ± 2.9	0.104	1, 12	0.753	19.1 ± 3.3	21.0 ± 2.3	0.193	1, 12	0.668
Web-builders	7.1 ± 1.4	8.8 ± 1.5	0.497	1, 12	0.494	7.6 ± 2.0	8.3 ± 1.0	0.073	1, 12	0.791
Hunters	12.2 ± 2.2	12.0 ± 1.6	0.008	1, 12	0.931	11.5 ± 31.7	12.8 ± 2.1	0.194	1, 12	0.668
Thomisidae	6.1 ± 2.0	6.4 ± 1.0	0.011	1, 12	0.918	5.9 ± 1.2	6.6 ± 1.9	0.099	1, 12	0.758
Gnaphosidae	1.0 ± 0.3	0.5 ± 0.3	1.714	1, 12	0.215	0.8 ± 0.3	0.8 ± 0.3	0.000	1, 12	1.000
Oxyopidae	2.0 ± 0.6	2.5 ± 0.7	0.289	1, 12	0.601	2.4 ± 0.8	2.1 ± 0.4	0.072	1, 12	0.793
Salticidae	3.0 ± 0.8	2.5 ± 0.7	0.211	1, 12	0.655	2.3 ± 0.8	3.3 ± 0.7	0.842	1, 12	0.377

## Data Availability

Data available upon request.

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
