# Peer review of "Side Effects of Kaolin and Bunch-Zone Leaf Removal on Generalist Predators in Vineyards"

_insects, 2023, doi:10.3390/insects14020126_

Round 1
Reviewer 1 Report
In general, the article is well structured, giving an interesting analysis on the impact of the application of kaolin and bunch-zone leaf removal on predators. This article can be an added value to support winegrowers on the implementation of IPM strategies, avoiding the use of insecticides.
Authors should clarify some minor aspects related with introduction, material and methods, results and discussion. Some comments are provided below:
Introduction /Discussion
Pg. 68-75 / Pg. 385-395 As the results of this article depend on the amount/frequency of the use of Kaolin (“a moderate use of kaolin has negligible impacts on generalist”), it might be relevant to quantify in the introduction / discussion the amount (dose) of Kaolin used in other studies.
In this study, the dose used was 2 kg/100 L (2% suspension). In other studies, related with the use of kaolin for sunburn protection, the maximum dose can reach 5 kg/100 L and that is a big difference in each treatment, with a probable higher impact on predators too…if possible, please add more information about amount of kaolin used in other studies, when compared with you results.
Material and methods / Results
Pg. 119-121 / Pg. 313-319. A significantly higher percentage of web-builders and salticidae were associated with yellow sticky traps than drop cloth method. According to these results, can you please clarify why yellow sticky traps were not performed in 2016? Moreover, can you add in discussion some information/recommendation about the best assessment method to implement in future studies related with yours?
Results
Pg. 183-184. The 1st point of results should be (3.1) Spiders and generalist predatory insects in vineyards and subpoints can be divided in 3.1.1 Spiders by drop cloth method and 3.1.2 Predatory arthropods by yellow sticky traps, following the organization of discussion section
Pg. 229. I suggest to remove “For the spiders collected with the drop cloth method in Vineyard A, as the previous paragraphs are also related with drop cloth method
Pg 259. Years instead of year
Pg. 291-292. Remove “studied only one year” as this was already explained on material & methods) and put only the year, following what was done in the other tables - “two Vineyards B (2015) and C (2016)
Pg. 304. Remove )
Reviewer 2 Report
The work is of considerable cognitive interest in the study of agronomic-cultural practices functional to the protection of the vineyard from arthropod pests in line with the principles of the IPM and the needs of sustainability.
The maintenance and, if possible, the enhancement of functional biodiversity is fundamental in the evolution of crop protection strategies. So, my warmest congratulations!
That said, I would like to know the following for any clarifications / integrations in the paper.
1) If and how the possible movement of leafhoppers and predatory flying insects, possibly induced by the application of kaolin towards the untreated plots, has been hypothesized / considered / examined.
2) If and how the possible predatory action of spiders against predatory insects has been considered / examined.
3) If and how the possible spatial competition within the spider community has been considered / examined.
Last but not least, I would like you to value / explain (perhaps in materials and methods) the importance of defining the periods of removal of the leaves from the bunch area and of the application of kaolin as an integrated function of the microclimatic and phytoiatric needs, these last primarily aimed at controlling Lobesia botrana, with particular reference to the period of hatching of the eggs of its carpophagous generations.
Apart from the comments above, corrections and changes to the text, in terms of proposals, are set out below and in the attached pdf of the manuscript.
Simple Summary
Line 11 – Consider replacing: “Integrated Pest Management” with: “Integrated Pest Management (IPM)”
Line 15 - Consider replacing: “negligible impacts” with: “a negligible impact”
Line 17 - Consider replacing: “Integrated Pest Management” with: “IPM”
Keywords
Line 30 - Sort by importance or alphabetically
Line 30 – Consider replacing: “natural enemies” with: “entomophagous arthropods”
Line 30 – Consider replacing: “predatory insects” with: “IPM”
Line 30 – Consider replacing: “diversity” with: “biodiversity”
1. Introduction
Lines 39 - Consider replacing: “pests’ control” with: “pest control”
Line 41 – Note the recent genus change and replace: “Empoasca vitis (Göthe)” with: “Hebata vitis (Göthe)” [Xu et al. (2021) - Phylogeny of the tribe Empoascini (Hemiptera: Cicadellidae: Typhlocybinae) based on morphological characteristics, with reclassification of the Empoasca generic group. Systematic Entomology, 46, 266–286 DOI: 10.1111/syen.12461]
Line 44 - Consider replacing: “(Denis and Schiffermüller)” with: “(Denis & Schiffermüller)”
Line 45 – With reference to “(Lepidoptera: Tortricidae)” taxonomically classify species in a homogeneous way by reporting order and family in brackets in all cases or in no case!
Line 51 – Replace: “newly-hatched larvae die” with: “newly-hatched larvae can die”
Lines 54-55 - Consider replacing: “with the added benefit of reducing berry sunburn damage often associated 54 with the latter practice alone” with: “with the further advantage, thanks to the kaolin, of reducing the possible damage to the grapes from sunburn”
Line 55 – Replace: “Integrated Pest Management” with: “Integrated Pest Management (IPM)”
Lines 56-57 – Replace: “synthetic insecticides” with: “pesticides”
Line 73 – Replace here and in all subsequent cases: “E. vitis” with: “H. vitis”
Line 78 – Replace: “this pruning strategy” with: “this cultural practice”
Line 82 – Replace: “the effects … was studied” with: “the effects … were studied”
2. Materials and Methods
Lines 107-108 – With reference to: “two times a year, except on Vineyard A in 2016 where three applications occurred because the first was notably washed off by rain” specify, in the text as well as in the supplementary file, not only the periods of the year but also the phenological stages of the vine and those of the carpophagous generations of Lobesia botrana in which the LR operations and the kaolin applications were carried out!
Line 139 - More than 35 species of the genus Aeolothrips Haliday, 1836 are recorded in Europe. Are you sure you only found one species? Is the abbreviation "sp." correct?
Lines 139-141 - Insert a colon after the names of the different orders (Thysanoptera: Aeolothripidae), (Coleoptera: Coccinellidae), (Neuroptera: Chrysopidae), and replace "Heteroptera" with "Hemiptera" (Hemiptera: Anthocoridae)
Line 141 - Fauna Europaea reports 24 species for the genus Orius Wolff, 1811. The use of the abbreviation "spp." is it appropriate for your data?
Line 171 - Consider replacing: “spiders’ species” with: “spider species”
3. Results
Line 234 – About “Orius spp.” see previous note about the abbreviation "spp."
Line 235 – About “Aeolothrips sp.” see previous note about the abbreviation "sp."
4. Discussion
Lines 325 – Replace “The Salticidae Salticus scenicus (Clerck)” with: “Salticus scenicus (Clerck) (Salticidae)
Line 329 – Replace: “The Oxyopidae O. lineatus” with: “Oxyopes lineatus (Oxyopidae)”
Line 333 – Replace: “E. fabae (Harris)” with: “Empoasca fabae (Harris)”
Lines 336-337 – Replace: “the Miturgidae Cheiracanthium spp. which was found” with: “Cheiracanthium spp. (Miturgidae), which were found”
Line 339 – Replace: “E. elegantula Osborn” with: “Erythroneura elegantula Osborn)”
Line 344 – Replace: “Araniella sp.” with: “Araniella sp. (Araneidae)”
Line 407 – Replace: “Anthocoris nemoralis (Fabricius)” with: “Anthocoris nemoralis (Fabricius) (Anthocoridae)”
Line 419 – Replace: “C. externa (Hagen)” with: “Chrysoperla externa (Hagen)”
5. Conclusions
Line 440 – Replace: “Integrated Pest Management (IPM)” with: “IPM”
Line 443 – With reference to “their decrease could be a consequence of the negative effects of kaolin on leafhoppers”, can you exclude that the difference in leafhopper population density between treated and untreated plots with kaolin could be enhanced by the possible transfer of leafhopper adults from treated to untreated parcels due to a deterrent effect of kaolin? What species and population densities of leafhoppers did you detect in the treated and untreated plots with kaolin?
References
Line 327 – Make sure that the citations in the text are correctly in order and duly appropriate. Also make sure that the editorial rules are perfectly respected.

Reviewer 3 Report
Dear Authors,
the text of your submitted paper Insects-2120088 needs to be improved particularly in some parts of the following chapters: Introduction. M & M, Results.The modifications required (adding of more data, re-writing of some parts, etc.) are important to ensure the scientific soundness of the article. Please, look at my notes in the word text attached below. In the Introduction, I suggest to re-write the periods , lines 49 to 52 and lines 55 to 58, also recording new references. Moreover, in regard to the aim of the experimental study, I suggest to focus ,firstly, the study of biodiversity in the selected area (vineyards) through the functional diversity indices you proposed and then the effects of the two methods, i.e. kaolin applications and LR. In M & M some data (i.e. dates of trials and samplings) have to be recorded in a short and clear way , not only in the Supplementary files, because some tables could not be understandable!) .I also suggest to re-write the period from line 161 to 176 because difficult to understand. I suggest, in the Results, to consider my notes for the period from line 184 to 189 and to re-write the caption of Figure 1, also reporting something about in the text. I strongly suggest a revision of the English text by a mother language lecturer.

Round 2
Reviewer 3 Report
Dear Authors ,
I read carefully the new version ( see the file peer-review-26187223.v1 attached below) and your replies to my 16 notes, in the word doc.insects-2120088-coverletter. I appreciated your effort to consider my suggestions and I thank you. In particular, I noted the improvement of several periods/phrases in the Introduction and M&M chapters, Tables and Figures captions in the Results and the revision of the References list. Also the review of the English language contributed to the good quality of text presentation.
Sincerely

Author Response
Thank you very much.
Best regards,
Francesco pavan